# Efficacy of Deep TMS with the H1 Coil for Anxious Depression

**DOI:** 10.3390/jcm11041015

**Published:** 2022-02-15

**Authors:** Gaby S. Pell, Tal Harmelech, Sam Zibman, Yiftach Roth, Aron Tendler, Abraham Zangen

**Affiliations:** 1BrainsWay Ltd., Jerusalem 9777518, Israel; tal.harmelech@brainsway.com (T.H.); sam@brainsway.com (S.Z.); yiftach@brainsway.com (Y.R.); aronte@brainsway.com (A.T.); 2Department of Life Sciences, Ben-Gurion University of the Negev, Be’er-Sheva 8410501, Israel; azangen@bgu.ac.il; 3Advanced Mental Health Care Inc., Royal Palm Beach, FL 33411, USA

**Keywords:** depression, anxiety, comorbid anxiety, anxious depression, transcranial magnetic stimulation, non-invasive brain stimulation

## Abstract

(1) Background: While the therapeutic efficacy of Transcranial Magnetic Stimulation (TMS) for major depressive disorder (MDD) is well established, less is known about the technique’s efficacy for treating comorbid anxiety. (2) Methods: Data were retrospectively analyzed from randomized controlled trials (RCTs) that used Deep TMS with the H1 Coil for MDD treatment. The primary endpoint was the difference relative to sham treatment following 4 weeks of stimulation. The effect size was compared to literature values for superficial TMS and medication treatments. (3) Results: In the pivotal RCT, active Deep TMS compared with sham treatment showed significantly larger improvements in anxiety score (effect size = 0.34, *p* = 0.03 (FDR)) which were sustained until 16 weeks (effect size = 0.35, *p* = 0.04). The pooled effect size between all the RCTs was 0.55, which compares favorably to alternative treatments. A direct comparison to Figure-8 Coil treatment indicated that treatment with the H1 Coil was significantly more effective (*p* = 0.042). In contrast to previously reported studies using superficial TMS and medication for which anxiety has been shown to be a negative predictor of effectiveness, higher baseline anxiety was found to be predictive of successful outcome for the H1-Coil treatment. (4) Conclusions: Deep TMS is effective in treating comorbid anxiety in MDD and, unlike alternative treatments, the outcome does not appear to be adversely affected by high baseline anxiety levels.

## 1. Introduction

Comorbid anxiety is common in patients with major depressive disorder (MDD), and these disorders may share a common underlying pathophysiology [1]. Approximately 50% of patients with MDD experience anxiety disorders during their lifetime, while up to 85% of patients with depression experience significant anxiety symptoms, and comorbid depression occurs in up to 90% of subjects with anxiety disorders [2,3,4]. Overall, lifetime comorbidity of the two syndromes is approximately 60% [5,6]. The most common Diagnostic and Statistical Manual of Mental Disorders (DSM-5) MDD specifier used is anxious distress, which is present in 75% of adults with major depressive disorder [7]. Various definitions of “anxious depression” have appeared in recent research, including MDD with a comorbid anxiety disorder, MDD with anxiety symptoms, and MDD with the anxious distress specifier as introduced in the DSM-5. Relying on ICD-10 or DSM diagnoses (i.e., syndromal criteria) yields a clinical picture of a comparatively mild or transient disorder. Using dimensional criteria such as DSM combined with additional rating scales—most commonly, the Anxiety/Somatization factor score from the Hamilton Depression Rating Scale (HDRS-A/S) [8]—yields a more serious clinical picture. The latter have, thus far, provided the most clinically relevant data on differences between subjects with anxious depression and those with depression or anxiety alone [9]. High prevalence rates of anxious depression (40–80%) are found regardless of the definition and assessment tools used [10]. Patients with anxious depression have worse clinical course trajectories and outcomes, including greater depression severity, lower remission rates, higher dysfunction, and increased risk for suicide [11].

Deep Transcranial Magnetic Stimulation (Deep TMS) treatment with the H1 Coil provides a safe and effective FDA-cleared clinical tool for the treatment of MDD in adult patients who did not achieve satisfactory improvement from earlier antidepressant medication treatment in the current episode. Since the original FDA (Food and Drug Administration) clearance of the H1 Coil for MDD in 2013 (510(k) No. K122288), Deep TMS has also been cleared for the treatment of other mental disorders, including the H7 Coil for the treatment of obsessive-compulsive disorder (OCD) and the H4 Coil for smoking addiction. Recently, the approval for comorbid anxiety in MDD was obtained (510(k) No. K210201). Here, we present the evidence on which this approval was based. This consists of retrospective analysis of the data from the multicenter pivotal randomized controlled trial (RCT) in combination with data from other RCTs that used the H1 Coil for treatment of MDD.

## 2. Materials and Methods

Across the collated RCTs, the stimulation protocol was the FDA-accepted high-frequency protocol for MDD treatment, i.e., 18 Hz, 2 s stimulation trains, 20 s inter-train intervals. The H1 Coil was placed on the left dorsolateral prefrontal cortex (dlPFC) following a 6 cm anterior shift from the position of hand motor threshold. The per-protocol (PP) data set was used for the main analyses. The primary measure was the adjusted Hamilton Depression Rating Scale (HDRS) Anxiety/Somatization (HDRS-A/S) factor score at baseline after 4 weeks of treatment. This sub-scale is commonly used to evaluate anxious depression [8,12] and is derived from the HDRS rating scale as the average total score of items 10, 11, 12, 13, 15, and 17. The adjusted model was based on mixed-model analysis using least-squares means (LSM). Within-group and between-group changes in this score can be evaluated. Within-group differences are defined as the treatment-induced change (i.e., baseline–endpoint). The analysis in this report focuses on the between-group differences, Δ, defined as the change from the start to the end of the treatment between treatment arms (i.e., real–sham). Between-group effect sizes (E.S.) were calculated using Cohen’s d and are shown with their 95% confidence interval limits (i.e., mean [lower bound, higher bound]).

In cases where the analysis of anxiety was secondary to the primary goal of the trial (i.e., the evaluation of anti-depressive effectiveness), statistical results were corrected for multiple comparison using the false discovery rate (FDR) approach.

The proportion of subjects who presented with anxious depression was calculated by employing the commonly used threshold at baseline of HDRS-A/S ≥ 7 as a marker of the syndrome [12]. Response rates in rating scales were defined by a treatment-induced decline of ≥50% of the scale at the treatment endpoint in PP completers. Comparisons were performed with the Chi-square test or Fisher’s exact test, as appropriate.

Additional analyses of secondary endpoints were performed for the pivotal multicenter RCT [13]. This included the analysis of changes in the Hamilton Anxiety Rating Scale (HAM-A) that enabled the selection of a treatment group with higher baseline levels of anxiety that is expected to be especially difficult to treat [14,15]. Since a mild-moderate HAM-A score is defined as a range between 18 and 24 points [16], the midpoint score of 21 was chosen as a cutoff to separate the subjects into groups with “low” and “high” levels of baseline anxiety. Logistic regression was also employed to find potential predictors of HAM-A and HDRS response with baseline HDRS and HAM-A values as dependent variables and the analysis applied separately for each treatment arm. Demographic variables were also included in the logistic model (age, gender) as well as the degree of treatment resistance (defined by Antidepressant Treatment History Form). The significant terms in the logistic regressions are described in the results as the model estimator together with their exponential (i.e., the odds ratio).

The expanded design of the pivotal RCT with 12 weeks of maintenance enabled the evaluation of the durability of the response in anxiety ratings until the study endpoint at 16 weeks. In case of missing values, data were imputed using the last observation carried forward (LOCF). Difference in adjusted rating scores between treatment arms at the study endpoints are reported with standard errors.

Following the evaluation of the pivotal multicenter data, this data were combined with two other RCTs that also used the H1 Coil to treat depression (see Table 1)—an independent head-to-head study comparing H1 Coil to Figure-8 Coil [17] and a trial carried out in subjects with late-life depression (LLD) [18]. The stimulation protocols were identical except for the number of daily pulses for the LLD study, which employed an extended protocol of 6012 pulses/day, and in the medication-free status of the pivotal multicenter trial. The data from these studies were collated, and a meta-analysis was performed to obtain an overall effect size.

In order to provide a basis for comparison of this value to alternative treatments, effect size measures were extracted from published studies that used the Figure-8 Coil for treatment of MDD [19,20]. In addition, studies of anxiolytic medication were collated for meta-analysis. This was carried out separately for investigations of anxious depression and generalized anxiety disorder (GAD). For the former, publications were selected based on a literature search performed using the search terms (Anxiety) AND (Somatization) AND (Depression); for the latter, data from trials in successful FDA New Drug Applications (NDAs) for the treatment of GAD were collected.

## 3. Results

### 3.1. RCTs That Used the H1 Coil

#### 3.1.1. Pivotal Multicenter RCT

The mean adjusted HDRS-A/S factor score showed a within-group decrease over the 4-week treatment period of 2.93 and 2.03 points for the H1 Coil and sham treatment arms, respectively (Figure 1). There was a statistically significant reduction in the adjusted HDRS-A/S factor score from baseline to primary endpoint between the treatment groups (i.e., between-group difference, Δ = 0.90 [95% CI: 0.08, 1.72]; *p* = 0.03 following FDR correction). The between-group effect size was 0.34 [95% CI: 0.01, 0.67] which represents a moderate effect size.

**Baseline anxiety**: When isolating the subjects with “high” levels of baseline anxiety based on the pre-selected HAM-A cutoff, the unadjusted between-group (i.e., H1 Coil vs. sham) difference (Δ = 4.52 [0.55, 8.49]) was found to be statistically significant (*p* = 0.036) with an effect size of 0.65 [0.07, 1.20], representing a medium effect. The corresponding group differences in the HDRS-21 rating scores were also significant (Δ = 5.10 [1.25, 8.95], *p* = 0.009) with an effect size of 0.75 [0.17, 1.31], representing a medium effect. This can be contrasted with values obtained for the group of subjects with “low” levels of baseline comorbid anxiety, for which neither HAM-A nor HDRS-21 rating scales showed a significant between-group difference (HAM-A rating scale: Δ = 0.76 [−1.20, 2.72] points, *p* = 0.61, E.S. = 0.16 [−0.25, 0.56]; HDRS-21 rating scale: Δ = 0.48 [−2.18, 3.14] points, *p* = 0.70, E.S. = 0.07 [−0.33, 0.48]). See Figure 2.

Logistic regression indicated the importance of pre-existing anxiety levels in the prediction of treatment response. For the outcome of HDRS response rates, lower baseline levels of HDRS and higher baseline levels of HAM-A were predictive of response (HAM-A baseline: Estimator = 0.117, exp(Estimator) = 1.124, *p* = 0.023; HDRS baseline: Estimator = −0.192, exp(Estimator) = 0.826, *p* = 0.018). For the outcome of HAM-A response, higher baseline levels of HAM-A were predictive of the response (HAM-A baseline: Estimator = 0.145, exp(Estimator) = 1.16, *p* = 0.049). The analysis thus implied that holding the other variables in the model constant and for a unit increase in the baseline HAM-A score, the odds for depression response increased by 12.4% (95% CI [2%, 24%]), while the odds for anxiety response increased by 15.6% (95% CI [0%, 34%]. By way of contrast, no model terms were significant for the same analysis applied to the sham treatment arm. Demographic and treatment resistance variables were not significant for any analysis.

This analysis thereby confirmed that the treatment of anxiety and depression symptoms was more effective in the set of subjects with more severe levels of pre-existing anxiety and, indeed, this patient characteristic was predictive of the treatment response.

**Durability**: The change in the adjusted HDRS-A/S factor score from baseline was maintained in the treatment arm until the final study endpoint at 16 weeks. A significant between-group difference of 0.92 [0.15, 1.69] points (*p* = 0.04 following FDR correction) was observed (Figure 3). This corresponded to a medium effect size of 0.35 [0.06, 0.65].

#### 3.1.2. Other RCTs

For the independent head-to-head trial [17] that directly compared H1 Coil to Figure-8 Coil treatments using an active control arm, adjusted between-group changes of HDRS-A/S in both coil groups (i.e., H1 Coil and Figure-8 arms) versus standard-of-care treatment (SOC) with medication were significant (H1 Coil vs. SOC: Δ = 2.46 [1.73, 3.19], *p* < 0.001 following FDR correction, E.S. = 0.91 [0.55, 1.26]; Figure-8 vs. SOC: Δ = 1.70 [0.99, 2.41], *p* < 0.05 following FDR correction, E.S. = 0.60 [0.27, 0.94)]. The boost in the effect sizes relative to the corresponding multicenter RCT values might be expected due to the use of an active control arm with medication. The study may in fact be more comparable to real-life treatment, in which the common options available to patients suffering from anxiety symptoms in MDD are treatments in the form of pharmacotherapy.

Importantly, the direct comparison of H1 Coil to Figure-8 Coil treatments of Δ = 0.76 [0.03, 1.49] points displayed a significant difference (*p* = 0.042; Figure 4). This result demonstrates that the Deep TMS treatment appears to be more effective for treating comorbid anxiety than Figure-8 Coil treatment.

For the late-life depression trial [18], the between-group adjusted difference in HDRS-A/S scores between H1 Coil and sham treatment arms showed a non-significant preference for the H1 Coil (Δ = 1.0 [0.2, 1.8], *p* = 0.22, E.S. = 0.36 [−0.23, 0.94]).

#### 3.1.3. Pooled Result

Across the RCTs, 48% of the subjects presented with anxious depression (see Table 1). For subjects treated with the H1 Coil, the anxiety response rates (defined using the HDRS-A/S scale) were 52% and 45% in subjects presenting with and without anxious depression, respectively (a non-significant difference). The pooled response rate in the H1 Coil treatment arm of 49% can be compared to the response rate of 28% in the Sham arm (*p* < 0.001).

No overall significant heterogeneity was found between the RCTs. The weighted adjusted between-group effect size of these three studies was 0.55 [0.15, 0.96], representing a medium effect size (Figure 5). If the head-to-head data were excluded from the pooling due to the use of an active control, the effect size was 0.34 [0.08, 0.6].

### 3.2. Comparison to Other Treatments

#### 3.2.1. Superficial TMS (Figure-8 Coil)

Among the many open-label trials that have used the Figure-8 Coil for the treatment of MDD, several have reported changes in anxiety scores that have generally supported its efficacy in treating anxiety symptoms associated with depression. These studies have used a variety of stimulation protocols reflecting the on-going search for the ideal protocol for the treatment of MDD with TMS. These protocols include high frequency-left dlPFC [21,22], low-frequency right dlPFC [23], and bilateral [24,25]. However, the clinical value of these studies is reduced by the lack of placebo arms. The small number of RCTs that have reported anxiety data include the Neuronetics registration trial [19], for which values of the HDRS-A/S scale were recorded in the study’s accompanying documentation [26]. From this source, an effect size could be calculated as 0.12 [−0.11, 0.34], which is considered as a small effect size. Access to the data set of the independent OPT-TMS trial which used the same Figure-8 Coil [20] was granted by the study’s principal investigator. The same adjusted model as performed in the H1 Coil analysis was applied to the OPT-TMS data until the end of the third week of treatment, following which the study design became adaptive. Using this approach, effect sizes of 0.20 [−0.10, 0.52] and 0.29 [−0.05, 0.58] were obtained for the HDRS-A/S scale for PP completers (*n* = 154) and modified intention-to-treat (mITT) (*n* = 190) data sets, respectively. Furthermore, the mixed-model analysis showed no significance of the Group × Time interaction term in either dataset, indicating that the effect of the real stimulation on the HDRS-A/S sub-scale over time using the Figure-8 Coil was not significantly different from the effect in the Sham arm. This can be contrasted to the results reported for the pivotal multicenter RCT with the H1 Coil, where the interaction term was statistically significant for the PP group (*p* = 0.008) [13].

It can therefore be concluded that the effect size obtained with the H1 Coil compares favorably with interventions using Figure-8 Coils for the treatment of comorbid anxiety in MDD.

#### 3.2.2. Anxiolytic Medication (Meta-Analyses)

**Anxious depression**: A comprehensive literature search for medication trials reporting the HDRS-A/S scale during treatment of comorbid anxiety in MDD returned the 14 studies summarized in Appendix A. The between-group difference (real vs. placebo) was Δ = 0.82 [0.60, 1.04]. The weighted average effect size was 0.27 [0.20, 0.34], which can be compared to corresponding values obtained with the H1 Coil (Section 3.1.3). It may be concluded that the Deep TMS treatment of comorbid anxiety in MDD offered results that are similar to or better than those of anxiolytic drugs for the treatment of this condition.

**Generalized anxiety disorder**: Trials used by the FDA for the approval of pharmacotherapy for GAD were collected and summarized in Appendix A. It should be noted that these trials relied on the HAM-A scale rather than the HDRS-A/S scale, since the latter is not commonly used for the rating of primary anxiety disorders. The between-group difference (real vs. placebo) was Δ = 2.56 [2.08, 3.04]. The overall weighted effect size was 0.31 [0.25, 0.37] (Appendix A). These values can be compared to those obtained for the H1 Coil treatment of comorbid anxiety (Section 3.1.3), in particular, to the effect size of 0.65 [0.07, 1.20] obtained using the same HAM-A scale for the subjects selected according to high levels of baseline anxiety in the multicenter RCT. While primary anxiety disorders such as GAD are not the focus of this report, these results are nevertheless illustrative in establishing what should be the ceiling of expected efficacy for anxiolytic medications in treating primary anxiolytic disorders rather than comorbid secondary anxiety.

Figure 6 summarizes the range of effect sizes collated in this report.

## 4. Discussion

The evaluation of the data collected from three RCTs of H1 Coil treatment of MDD indicated a significantly improved anxiolytic effect relative to sham treatments, with rapid onset. The analysis of the multicenter RCT data set demonstrated sustained durability up to 16 weeks. The head-to-head trial indicated the significantly improved effectiveness of Deep TMS treatment over Figure-8 Coil treatment for the treatment of comorbid anxiety. The pooled data also showed a moderate effect size compared to sham of 0.55, or 0.34 with the exclusion of the trial that used an active control. Results were also evaluated for comparison from collation of pharmacotherapeutic trials. The values indicate a clinically meaningful response in comparison to equivalent trials with superficial TMS and anxiolytic medication which are equivalent to or better than these existing treatments.

The results in this report confirm previous attempts to evaluate the influence of anxiety on H1 Coil treatment of MDD that were reported in two meta-analyses [27,28]. Both studies pooled mainly open-label studies and reported a significant within-group effect in the H1 Coil treatment arm (Hung et al., 2020: pooled weighted E.S. (Hedge’s g) = −1.282 [−1.514, −1.051], *p* < 0.001 [27]; Kedzior et al., 2015: pooled weighted E.S. (Cohen’s d) = 1.45 [1.10, 1.80]; *p* < 0.001 [28]).

The finding that baseline levels of anxiety are a positive predictor of the effectiveness of Deep TMS treatment stands in contrast to findings with either pharmacotherapies or TMS treatment with Figure-8 Coils. In both cases, baseline levels of anxiety were found to be a negative predictor of treatment success. For example, Lisanby et al. showed that the performance of the Figure-8 Coil treatment of MDD was poorer in the subjects with comorbid anxiety disorders [14]. Similarly, the recent “THREE-D study” comparing intermittent theta burst with rTMS treatments using Figure-8 Coils reported that the performance of the depression treatment was poorer in patients who started the treatment with anxiety symptoms [15]. For pharmacotherapy, a poorer response to antidepressants in patients with anxiety was a key finding of the STAR*D trial [12,29]. This finding has been confirmed in subsequent trials [30,31,32], although there have been inconsistent results [33].

It is therefore a highly significant finding in this analysis that baseline levels of anxiety are a positive rather than a negative predictor of the efficacy of Deep TMS treatment of depression and anxiety symptoms in MDD. This is suggestive of a unique ability of the H1 Coil relative to other treatments to treat anxiety comorbid to depression. These results correspond to our recent work suggesting the distinctive ability of Deep TMS to treat a wide range of symptoms of MDD, that included core depression and anxiety [34,35]. This capability of performing a so-called “polysymptomatic” treatment with Deep TMS is an area of on-going investigation.

There are several limitations to our work that should be taken into account when interpreting the findings. These include the retrospective nature of the analysis presented in this report. The trials were all designed to treat major depression as a primary objective, with anxiety as a secondary endpoint. Correction for multiple correction was used to statistically mitigate this. However, a trial designed with the primary goal of investigating comorbid anxiety in MDD would clearly represent the optimal choice to investigate the treatment. The poolability of RCTs using the H1 Coil may be adversely affected by differences in the use of concurrent medication and also by the inclusion of the head-to-head trial with its use of an active (medication) control. The pooled value with the exclusion of this data set was therefore also provided for comparison.

The HDRS Anxiety/Somatization (HDRS-A/S) score was the main rating scale used to assess anxiety changes in the analysis. As a sub-scale derived from the HDRS rating scale, it may not be as rich as a specialized rating scale such as the HAM-A scale. However, the sub-scale approach is an efficient way to collect symptom-specific data, and the widespread uptake of the HDRS-A/S scale is reflected by its accepted use to define anxious depression. On the other hand, the field of anxiety treatment has been hampered by the lack of a concrete definition of what constitutes anxious depression. It has been proposed that anxious depression can be defined as a diagnosis of depression with subthreshold anxiety symptoms, while comorbid anxiety represents a joint diagnosis of depression and anxiety [36]. This was reflected in our analysis with the choice of a more conservative anxiety scale threshold based on the HAM-A scale for the definition of a group with substantial levels of baseline comorbid anxiety.

Another limitation is the focus on data in this report from clinical trials, which may limit the extendibility of the results to real-life settings. On-going data collection with Deep TMS is currently underway as part of a registry which will provide data from a naturalistic setting.

## 5. Conclusions

We conclude that Deep TMS with the H1 Coil is an effective and robust treatment for anxious depression, a challenging and frequently refractory condition. Unlike alternative treatments, the outcome does not appear to be adversely affected by high baseline anxiety levels, which has been a limiting factor with existing treatments such as superficial TMS and anxiolytic medication.

## Figures and Tables

**Figure 1 jcm-11-01015-f001:**
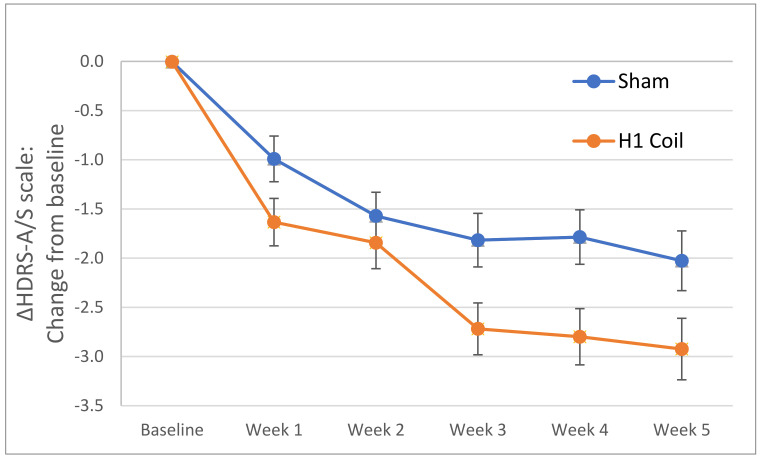
Change (i.e., Δ) in HDRS Anxiety/Somatization (HDRS-A/S) score from baseline to the primary study endpoint (week 5) for the H1 Coil and Sham arms in the per-protocol analysis (pivotal multicenter RCT [13]). The positive influence of the H1 Coil treatment on the anxiety score is seen to develop from the first post-baseline measurement, and the performance is consistently improved over the Sham arm. Error bars are ± standard errors in the mean (SEM).

**Figure 2 jcm-11-01015-f002:**
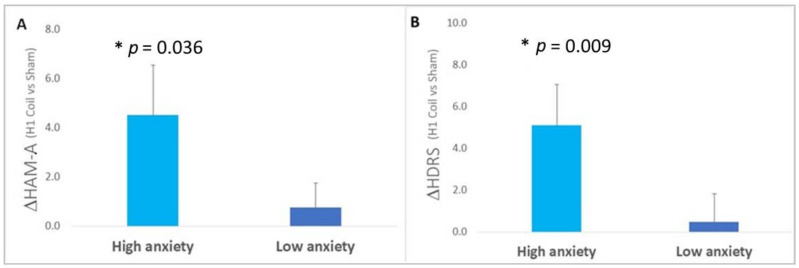
The influence of baseline anxiety symptoms on treatment outcome in the pivotal RCT [13]. Between-group (i.e., H1 Coil–Sham) differences in anxiety and depression outcomes during the treatment, with the subjects split into two sub-groups selected to reflect differences in the severity of anxiety at baseline (i.e., Low Anxiety sub-group: baseline HAM-A < 21 points, High Anxiety sub-group: baseline HAM-A ≥ 21 points). Between-group changes in (**A**) HAM-A anxiety score and (**B**) HDRS-21 overall depression score until the primary study endpoint are shown (i.e., baseline–endpoint indicated by Δ). A significant improvement in the performance of the H1 Coil relative to Sham for both measures was found in the sub-group with high anxiety at baseline. Error bars are ± SEM. * indicates a significant between-group difference (*p* < 0.05).

**Figure 3 jcm-11-01015-f003:**
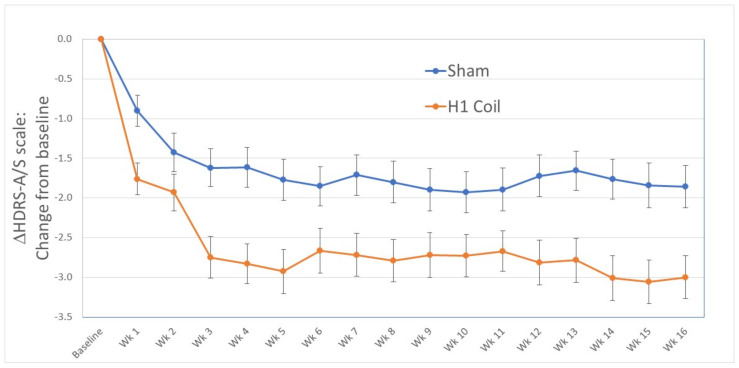
Between-group change in HDRS Anxiety/Somatization (HDRS-A/S) score from baseline to the final rating at week 16 for the H1 Coil and the Sham arms in the per-protocol analysis (pivotal multi-center RCT [13]). Daily treatments until the end of week 4 were followed by bi-weekly treatments for 12 weeks. Missing values were corrected with LOCF imputation. The treatment effect is shown to be sustained from the primary study endpoint until the final rating at week 16. Error bars are ± standard errors in the mean (SEM).

**Figure 4 jcm-11-01015-f004:**
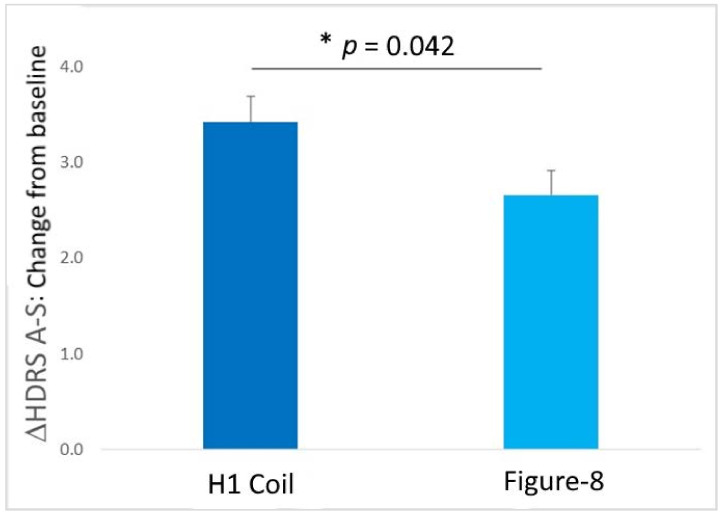
Within-group (i.e., Baseline–Endpoint) comparison of changes in HDRS-A/S rating scale over the course of the treatment in the head-to-head clinical trial [17] for the two TMS coil treatment arms (H1 Coil and Figure-8). The results indicate a significant difference in the treatment-induced changes in anxiety-related symptoms, with the improvement following H1 Coil treatment significantly larger than the improvement following Figure-8 Coil treatment (*p* = 0.042). Error bars are ± SEM. * indicates a significant between-group difference (*p* < 0.05).

**Figure 5 jcm-11-01015-f005:**
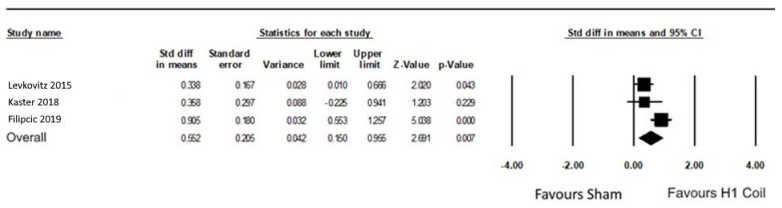
Forest plot displaying the randomized controlled trials that used the H1 Coil to treat comorbid anxiety in MDD ([13,17,18]). Anxiety ratings were based on the HDRS-A/S factor score.

**Figure 6 jcm-11-01015-f006:**
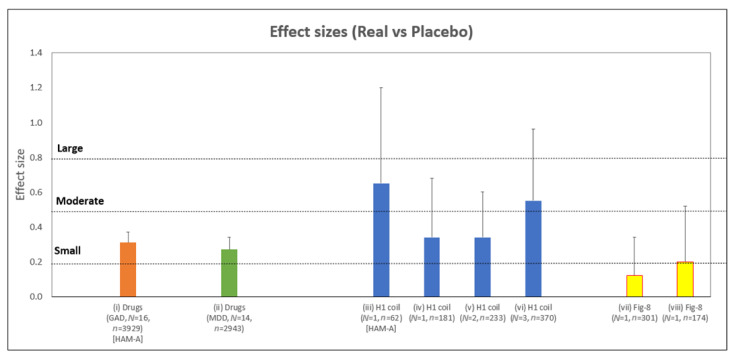
Summary of between-group (i.e., real vs. placebo) effect sizes described in this report that evaluate the changes of anxiety symptoms in primary (GAD) and secondary (MDD) disorders relative to placebo treatment. All values are based on the HDRS-A/S factor scale and concern the treatment of MDD, unless otherwise indicated. Studies are as follows (left-right): (i) Medications in GAD RCTs based on HAMA-A rating scale (Section 3.2.2, also Appendix A); (ii) Medications in anxious depression RCTs (Section 3.2.2, also Appendix A); (iii) H1 Coil data from the pivotal MDD RCT [13] based on the HAMA-A rating scale for the sub-group with high anxiety at baseline defined by HAM-A ≥ 21 (Section 3.1.1); (iv) H1 Coil data from pivotal MDD RCT (Section 3.1.1); (v) Two MDD RCTs using H1 Coil (Section 3.1.3) [13,18]; (vi) Three MDD RCTs using H1 Coil (Section 3.1.3 and Figure 5) [13,17,18]; (vii) Figure-8 Coil data from Neuronetics registration MDD RCT [19] (Section 3.2.1); (viii) Figure-8 Coil data from OPT-TMS MDD RCT [20] (Section 3.2.1). Error bars indicate the extent of the upper bound of the 95% confidence interval. The accepted limits of small (0.2), moderate (0.5), and large (0.8) effect sizes for Cohen’s d are indicated (horizontal dotted lines). *N* = number of trials. *n* = number of subjects.

**Table 1 jcm-11-01015-t001:** Summary of RCTs using the H1 Coil for the treatment of MDD, evaluated for the influence on comorbid anxiety.

Trial *	Population	Sample Size (N), Age (SEM)	Protocol	Primary Endpoint	Secondary Endpoints	Within-Group Effect Size **	% Anxious Depression ***
Levkovitz 2015 [13]	Treatment-Resistant Depression, tapered off all antidepressants, mood stabilizers, and antipsychotics in wash-out period (1–2 weeks) prior to trial	N:	18 Hz, 120% MT, 20 min, 1980 pulses/session 20 sessions (4 weeks) followed by maintenance (2 sessions/week for 12 weeks)	Change in HDRS at 5 weeks	Change in HDRS- A/S at 5 weeks. Change in HAM-A at 5 weeks for patients with baseline HAM-A ≥ 21		61%
		H1: 89				1.16	
		Sham:92				[0.85, 1.48]	
		Age:					
		H1: 45.1 ± 11.7					
		Sham: 47.6 ± 11.6					
Kaster 2018 [18]	Treatment-Resistant Late-Life Depression (LLD), continued psychotropic medications unchanged for trial duration	N:	18 Hz, 120% MT, 61 min, 6012 pulses/session 20 sessions (4 weeks)	HDRS Remission at 4 weeks	Change in HDRS- A/S at endpoint		60%
		H1: 25				1.15	
		Sham: 27				[0.47, 1.82]	
		Age:					
		H1: 65.0 ± 5.5					
		Sham: 65.4 ± 5.5					
Filipčić 2019 [17]	Treatment-Resistant Depression, on standard pharmacotherapy unchanged for trial duration	N:	18 Hz, 120% MT, 20 min, 1980 pulses/session, 20 sessions (4 weeks)	HDRS Remission at 4 weeks	Change in HDRS- A/S at endpoint		35%
		H1: 65				1.63	
		Fig8: 72				[1.21, 2.06]	
		SOC: 72					
		Age:					
		H1: 50 ± 2.7					
		Fig8: 51 ± 2.7					
		SOC: 53 ± 2.2					

* The ClinicalTrials.gov identifiers for the RCTs are: NCT00927173 [13]; NCT02917499 [17]; NCT01860157 [18]. ** The adjusted within-group effect size summarizes the change from baseline to study endpoint at 4 weeks in the Real treatment arm. It is based on the adjusted least-squared means (LSM) analysis. *** The proportion of patients in the trial who presented with anxious depression (defined by HDRS-A/S ≥ 7). H1: H1 Coil; Fig8: Figure-8 coil. Ages are shown with standard error in the mean (SEM). SOC: Standard-of-care treatment (medication).

## Data Availability

The data presented in this study are available on request from the corresponding author. The data are not publicly available due to privacy restrictions.

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
