# Peer review of "Efficacy of Deep TMS with the H1 Coil for Anxious Depression"

_jcm, 2022, doi:10.3390/jcm11041015_

Round 1
Reviewer 1 Report
Reviewer comments for the manuscript: “Efficacy of Deep TMS with the H1 Coil for Anxious Depression” [Manuscript Number: J. Clin. Med. 2022, 11, x. https://doi.org/10.3390/xxxxx] By Pell et al.; Submitted for consideration in Journal of Clinical Medicine
The authors have attempted to study the effects produced by Deep TMS with H-coil on patients with anxious depression. For this, the authors performed a meta-analysis of Deep TMS with H-coil and compared that to sham coils.
The manuscript is well written and if authors address the following issues and modify their manuscript, it can be considered for publication as per the Editor’s discretion.
Major suggestions
- All the graphs could be made better with proper descriptions of X- and Y-axes and explaining the results clearly in the legend
- Can put a table of studies showing the less effectiveness of superficial TMS (figure of 8) in anxious depression compared to deep TMS
- Late-life depression (LLD) study, which employed an extended protocol of 6012 pulses/day which is not matching with the pulse of other 2 studies - line no: 106, table no: 1 , 2nd study
- In table no: 1, 2nd & 3rd study in Treatment Resistant Depression, standard pharmacotherapy unchanged for the trial duration while the 1st study tapered off all antidepressants, mood stabilizers, and antipsychotics in wash-out period (1-2 weeks) prior to trial. Authors need to explain these differences in the discussion. Also authors need to clearly differentiate the meta-analysis part of the study.
Minor corrections (English proof-reading is suggested)
- The pooled effect size over all the RCTs was 0.55 - Overall - line no: 17
- including the H7 Coil for obsessive compulsive disorder (OCD)- the obsessive compulsive disorder- line no: 55
- and comparisons performed with the Chi-square test or Fisher’s exact test as appropriate- were performed- line no: 78
- In case of missing values, data was imputed using last observation carried forward- the last observation – line no: 96
- at the study end point are reported with standard errors- end points are- line no: 98
- The stimulation protocols were identical with the exception of the number of daily pulses- instead of that except for can be used – line: 106
- The data from these studies were collated and a meta-analysis performed to obtain an overall effect size- was performed- line : 108
- but the difference were not found not be significant- differences were/ difference was- line 216
- Response rates are summarized Table 2.- Response rates are summarized in Table 2- line: 218
- If the head-to-head data is excluded from the pooling due to its used of an active control,- due to its use of an active control . line no: 227
Author Response
We thank the reviewers for their comments.
Major suggestions
- All of the figures have been modified to improve their clarity in particular with regard to x and y axis titles. For example, y-axis titles that describe between-group differences in e.g., ΔHAMA-A, are now followed by a description (e.g., H1 Coil vs Sham). All figure legends have also been expanded to provide more information. For example, bar chart titles in Figure 6 which contains a considerable amount of information sources have been simplified and an expanded Figure legend clarified these sources.
- The use of figure-8 TMS to treat comorbid anxiety in depression has been largely limited to open-label trials. There are only a limited number of RCTs that used this coil, in particular the Neuronetics and OPT-TMS studies which are described in the report. The report is focused on between-group (i.e., real vs sham) rather the within-group (i.e., baseline - endpoint) effects. As a response to the reviewer's comment to discuss this literature, additional sentences have been added in the results section 3.2.1 (page 8, lines 241-246) in which the open-label literature is discussed.
- As the reviewer points out that the LLD study is unique among the 3 described RCTs in its use of 6012 pulses (rather than 1980 pulses). This point is explicitly mentioned in the text and in table 1. The reason for the increased number of pulses in this study is as a consequence of the expected highly refractory nature of depression in this age group. Whilst this logic is sound, there is little evidence that protocols with greater number of pulses are indeed more effective (for example, Fitzgerald 2019 [24]). For this reason, this factor is not believed to be a barrier in poolability between the studies.
- The reviewer importantly highlights the important differences in the medication-use between the studies. The key pivotal trials of rTMS in MDD isolated the effects of the stimulation treatment by tapering patients off medication before starting the treatment. In clinical practice however, it is acceptable and common practice that medications are not discontinued. Meta-analyses have shown moderate benefits of this strategy (for example, Sehatzadeh S et al.. J Psychiatr Neurosci. 2019 1;44(3):151) but the results are not equivocal and a recent review’s conclusion is merely that there “are no deleterious effect of concurrent medication use” (Fitzgerald P. J Affect Disord. 2020 1;276:90) with the exception of benzodiazepines. The influence of poolability on the different medication status of the trials was therefore not thought to be significant. A sentence regarding the issue of concurrent medication has now been added to the text. (page 10, line 356).
The sections describing results of meta-analyses have been better distinguished with an expanded Section Title (3.2.2 - Anxiolytic medication (meta-analyses)).
Minor corrections
All of these points have been corrected in accordance with the reviewer’s comments.
In preparing the report for submission, a number of minor corrections were made to the text. In particular, a line has been added to the introduction to correct the omission of the details of FDA 510(k) approval on which this report is based (page 2, line 57).

Reviewer 2 Report
The authors presented their findings from exploratory analyses from data from randomized controlled trials with deep TMS with the H1 coil for MDD for anxiety symptoms in depression. The authors reported interesting findings with improvements in anxiety in MDD, sustained until 16 weeks, potentially more significant than the improvements noticed with superficial rTMS, and surprisingly, higher levels of anxiety were positively related to the improvements, differently from what was observed in trials with superficial rTMS, potentially supporting the use of deep TMS for patients with MDD with high levels of anxiety, which tends to be highly prevalent in this population. The authors have reported on the limitations of this study, and the article is well written.
Author Response
We thank the reviewer. We cannot see any comments to answer from the reviewer.
Round 2
Reviewer 1 Report
I don't have any more comments, thanks